# Transcriptional Regulation and Functional Characterization of the Plasmid-Borne *oqxAB* Genes in *Salmonella* Typhimurium

Bill Kwan-wai Chan,[a,b] Marcus Ho-yin Wong,[a] Edward Wai-chi Chan,[a] Sheng Chen[b]

aState Key Laboratory of Chemical Biology and Drug Discovery, Department of Applied Biology and Chemical Technology, the Hong Kong Polytechnic University, Kowloon, Hong Kong

bDepartment of Infectious Diseases and Public Health, Jockey Club College of Veterinary Medicine and Life Sciences, City University of Hong Kong, Kowloon, Hong Kong

Bill Kwan-wai Chan and Marcus Ho-yin Wong contributed equally to this article. Author order was determined by the corresponding author after negotiation.

**ABSTRACT** Coexistence of *oqxAB* and *aac(6')-lb-cr* is often associated with the expression of fluoroquinolone resistance in *Salmonella*. The actual role of the plasmid-borne *oqxAB* gene and its regulatory mechanism compared to its chromosomally encoded counterpart in *Klebsiella pneumoniae* remain unclear We found that cloning of *oqxAB* gene only or chromosomally encoded *oqxABR* (ABRc) locus did not lead to an increase of ciprofloxacin (CIP) minimum inhibitory concentration (MIC) in *S*. Typhimurium, while cloning of the plasmid-encoded *oqxABR* (ABRp) locus led to a 4-fold increase in CIP MIC, reaching 0.0065 $\mu$g/mL. The co-carriage of these constructs with *aac(6')-lb-cr* further increased the CIP MIC to 0.25 $\mu$g/mL in *S*. Typhimurium carrying *aac(6')-lb-cr* and ABRp. Analysis of the transcription start site sequences showed that the expression level of suppressor protein gene, *oqxR*, in strains carrying ABRp was lower than that of its chromosomal counterpart due to the truncated promoter region in ABRp. The lower expression of OqxR in ABRp led to the overexpression of OqxAB, which elevated CIP MIC and exhibited a synergistic antimicrobial effect with the *aac(6')-lb-cr* gene product to confer intermediate CIP (MIC = 0.25 $\mu$g/mL) in *S*. Typhimurium. Global transcriptional regulators in *S*. Typhimurium did not seem to play a role in regulating the plasmid-borne *oqxAB* genes. In conclusion, findings in this work showed that neither *aac(6')-lb-cr* nor *oqxABRp*, but the combination of both genes, could mediate intermediated resistance to fluoroquinolone in *Salmonella*. The truncated promoter region in the *oqxR* gene of the plasmid-encoded locus led to the constituted expression of *oqxAB* genes.

**IMPORTANCE** The transferable mechanisms of quinolone resistance (TMQR) gene, *oqxAB*, has been widely detected in *Salmonella* and is commonly associated with *aac(6')-lb-cr*. It is thought to be associated with fluoroquinolone resistance, while its ancestor gene from *K. pneumoniae* is not. This study evaluated the actual role of the plasmid-borne *oqxAB* genes in *Salmonella* and showed that it was not able to mediate intermediated resistance to fluoroquinolone and only did so when it coexisted with *aac(6')-lb-cr*. Chromosomally encoded oqxABRc from *K. pneumoniae* was not able to mediate enhanced CIP MIC due to tight regulation by the suppressor *oqxR*. However, plasmid-encoded oqxABRp enabled *oqxAB* to be expressed constitutively due to the truncated promoter region of *oqxR*, leading to lower expression of the suppressor *oqxR*. This study clarified the roles of *oqxAB* and *aac(6')-lb-cr* in mediating fluoroquinolone resistance in *Salmonella* and provides insights into the regulation of plasmid-encoded TMQR determinant, *oqxAB*.

**KEYWORDS** *Salmonella*, OqxAB, TMQR, regulation, PMQR

Address correspondence to Sheng Chen, shechen@cityu.edu.hk.

The authors declare no conflict of interest.

Non-typhoidal *Salmonella* is the leading cause of foodborne illness worldwide, yet management of invasive infections caused by *S*. Typhimurium has become a major challenge due to the dissemination of multidrug-resistant strains (1). Recently, OqxAB, a

resistance-nodulation-division-type efflux pump, has emerged as a key transferable mechanism of quinolone resistance (TMQR) determinant among various members of the *Enterobacteriaceae* family, including *Salmonella* (2, 3). The pump was found to be one of the endogenous efflux systems in *Klebsiella pneumoniae* (4). Dissemination of the *oqxAB* gene is possibly due to acquisition of the entire locus from the chromosome of *K. pneumoniae* by IS*26* transposase, resulting in the formation of a transposon named Tn*6010*, within the *oqxA*, *oqxB* and *oqxR* genes are flanked by two IS*26* elements (5). This transposon was subsequently transposed into conjugative plasmids that circulate among Gram-negative bacterial pathogens. Resistance to olaquindox (substrate of OqxAB) and reduced susceptibility toward ciprofloxacin ($0.25 \leq$ MIC $< 1$ $\mu$g/mL) has been consistently observed in organisms harboring the Tn*6010*-borne *oqxAB* gene, which is often accompanied by *aac(6')-Ib-cr*, another common TMQR determinant (3, 6, 7). It remains uncertain whether this reduced fluoroquinolone susceptibility was due to *oqxAB*, *aac(6')-Ib-cr*, or the combined effect of both determinants. In *K. pneumoniae*, reduced ciprofloxacin susceptibility solely due to overexpression of *oqxAB* has only been recently established. The study showed that overexpressed *oqxAB* in *K. pneumoniae* increased the ciprofloxacin MIC from $\leq 0.064$ $\mu$g/mL to $> 3$ $\mu$g/mL (8). As an endogenous efflux gene, the expression of *oqxAB* in *K. pneumoniae* is regulated at both local and global levels, which is similar to the *acrAB* regulatory mechanism in other Gram-negative bacteria. Genetically, *oqxAB* in *K. pneumoniae* is flanked by two local transcriptional regulators, namely, *rarA*, and *oqxR*; the former is an activator for *oqxAB* whereas the latter is a repressor (9). Expression of *oqxAB* is also subject to regulation by the global regulator RamA (10). Unlike its chromosomal counterpart, *rarA* is missing from the Tn*6010*-*oqxAB* fragment, probably due to the excision process mediated by IS*26* (5). It is therefore unclear how this RND-efflux pump is regulated in *Salmonella*. It has been previously shown that the plasmid-borne *oqxAB* operon is constitutively expressed in *Salmonella* Typhimurium (4). While global regulators in *S.* Typhimurium, such as *ramA*, *marA*, and *soxS*, play an integral role in regulating the host's endogenous efflux gene *acrAB* (11), whether the plasmid-borne *oqxAB* genes are also subject to regulation by these elements and, if so, how such a regulatory mechanism is controlled by cellular signals from the host, remains to be elucidated. We hypothesized that the *oqxR* gene in the plasmid-borne *oqxABR* locus may not be fully functional, resulting in overexpression of *oqxAB*. Because it is an RND efflux pump, we also surmised that *oqxAB* may be subjected to global regulatory signals from the host. This study aims to evaluate the actual role of the plasmid-borne *oqxAB* genes in mediating changes in fluoroquinolone susceptibility, and to elucidate the mechanism by which expression of this efflux pump is regulated in *S.* Typhimurium.

## RESULTS

**OqxAB does not cause elevation of ciprofloxacin MIC against *S.* Typhimurium.** Although *oqxAB* is considered one of the TMQR genes, its functional role in contributing to the expression of fluoroquinolone resistance in *Salmonella* has not been confirmed. Previous studies showed that cloning the complete coding DNA sequence of *oqxAB* from both pHK06-53 and MGH78578 into a cloning vector, followed by transformation into *Escherichia coli*, only conferred a low minimum inhibitory concentration (MIC) of 0.008 $\mu$g/mL (12), suggesting that *oqxAB* is not a main determinant of fluoroquinolone resistance in *Salmonella*, although it is a typical TMQR gene on its own. In this study, we cloned the full length of *oqxA* and *oqxB* into the pACYCDuet-AmpR vector to create p*AB*, followed by transformation into *S.* Typhimurium PY1. The ciprofloxacin (CIP) MIC of the transformant was 2-fold lower than that of the vector control in the *S.* Typhimurium PY1 strain, implying that *oqxAB* did not reduce ciprofloxacin susceptibility in *Salmonella* (Table 1). In *Salmonella*, *oqxAB* is commonly associated with a mobile element, Tn*6010*, and exists in the form IS*26*-*oqxA*-*oqxB*-*oqxR*-IS*26*. Its original sequence from the *K. pneumoniae* chromosome was identical to the one located in the plasmid, except that the sequence upstream of the open reading frame of *oqxR* in the *oqxABR* locus in the plasmid was truncated by the IS*26* element. Unlike the

**TABLE 1** MICs of different antimicrobials tested on strains carrying different constructs[a]

| Strain | Constructs | MIC (µg/mL) | | | | | |
|---|---|---|---|---|---|---|---|
| | | CIP | GEN | NAL | OLA | CTX | CHL |
| PY1 | VC | 0.0156 | 1 | 4 | 8 | ≤0.0625 | 4 |
| PY1 | AB | 0.0078 | 2 | 4 | 16 | ≤0.0625 | 4 |
| PY1 | ABRp | 0.0625 | 2 | 16 | 64 | ≤0.0625 | 32 |
| PY1 | ABRc | 0.0156 | 0.5 | 2 | 8 | ≤0.0625 | 4 |
| PY1 | AAC | 0.0312 | 2 | 4 | 16 | ≤0.0625 | 4 |
| PY1 | AAC-AB | 0.0312 | 0.5 | 2 | 8 | ≤0.0625 | 4 |
| PY1 | AAC-ABRp | 0.25 | 2 | 16 | 64 | ≤0.0625 | 32 |
| PY1 | AAC-ABRc | 0.0625 | 2 | 8 | 16 | ≤0.0625 | 16 |
| PY1Δ*ramA* | VC | 0.0156 | 1 | 4 | 8 | ≤0.0625 | 4 |
| PY1Δ*ramA* | AAC | 0.0312 | 2 | 2 | 8 | ≤0.0625 | 4 |
| PY1Δ*ramA* | AAC-AB | 0.0312 | 0.5 | 2 | 8 | ≤0.0625 | 4 |
| PY1Δ*ramA* | AAC-ABRp | 0.25 | 1 | 16 | 64 | ≤0.0625 | 16 |
| PY1Δ*ramA* | AAC-ABRc | 0.0625 | 2 | 4 | 16 | ≤0.0625 | 16 |
| PY1Δ*soxS* | VC | 0.0156 | 1 | 4 | 8 | ≤0.0625 | 4 |
| PY1Δ*soxS* | AAC | 0.0312 | 2 | 4 | 32 | ≤0.0625 | 8 |
| PY1Δ*soxS* | AAC-AB | 0.0312 | 0.5 | 2 | 8 | ≤0.0625 | 4 |
| PY1Δ*soxS* | AAC-ABRp | 0.25 | 1 | 16 | 128 | ≤0.0625 | 16 |
| PY1Δ*soxS* | AAC-ABRc | 0.0625 | 2 | 4 | 16 | ≤0.0625 | 16 |
| ATCC 25922 | | 0.0078 | 2 | 4 | 8 | ≤0.0625 | 4 |

[a]ATCC 25922, the *E. coli* reference strain, was included as a quality control in antimicrobial susceptibility tests; PY1, *S.* Typhimurium PY1; VC, vector control; AAC, construct carrying *aac(6′)-ib-cr*; AB, constructs carrying *oqxAB*; ABRp, constructs carrying *oqxAB*, and the *oqxR* regulator gene originating from pHK06-53 (GenBank ID: KT334335); ABRc, construct carrying *oqxAB* and the *oqxR* regulator gene originated from *K. pneumoniae* MGH78578; CIP, ciprofloxacin; GEN, gentamicin; NAL, nalidixic acid; OLA, olaquindox; CTX, cefotaxime; CHL, chloramphenicol.

chromosome of *K. pneumoniae*, which contains a ~400-bp sequence upstream of *oqxR* ahead of the next open reading frame (ORF), an intergenic region of only 100 bp was detectable between *oqxR* and IS*26* in Tn*6010*, within which only 50 bp adjacent to *oqxR* are identical to those of the chromosomal sequence (Fig. 1a). We next generated two constructs to include the full length of *oqxABR* sequences, namely, pABRp and pABRc, that covered the entire *oqxABR* locus of plasmid and chromosomal origin, respectively. The CIP MIC for *S.* Typhimurium carrying pABRp was found to be 0.0625 µg/mL, 4-fold higher than that of the vector control. In contrast, the CIP MIC for the construct pABRc was 0.0156 µg/mL, the same as that of the vector control. Such results indicating that even the Tn*6010*-borne *oqxAB* could only slightly elevate the CIP MIC in *Salmonella* Typhimurium (Table 1).

**Synergistic effect of *aac(6′)-Ib-cr* and *oqxAB* on mediating ciprofloxacin resistance in *Salmonella*.** The slight decrease in ciprofloxacin susceptibility due to acquisition of *oqxAB* in *Salmonella* as observed above did not corroborate with the significantly elevated CIP MIC of *Salmonella* Typhimurium field strains, which normally exhibit CIP MIC in the range of 0.25 to 1 µg/mL if the strains only consisted of *oqxAB* but without others resistance determinants (Table S3 in the supplemental material). We then tested if the elevated CIP MIC was contributed by *aac(6′)-Ib-cr*, a common neighboring gene of the plasmid-borne *oqxAB* gene, rather than by *oqxAB* itself. We cloned *aac(6′)-Ib-cr* gene originated from pHK06-53 into a cloning vector, pACYCDuet-AmpR, along with 213 bp upstream of *aac(6′)-Ib-cr*, to create pAAC(6′)-Ib-cr. The CIP MIC of *S.* Typhimurium PY1 carrying this construct was 0.0312 µg/mL, i.e., only a 2-fold increase compared with that of the vector control, indicating that *aac(6′)-Ib-cr* itself did not actually confer intermediate resistance (MIC ≥ 0.5 µg/mL) to ciprofloxacin. Such a phenomenon has also been reported in previous studies (13, 14). *S.* Typhimurium PY1 carrying pAAC-AB, pAAC-ABRp, and pAAC-ABRc exhibited CIP MICs of 0.0312, 0.25, and 0.0625 µg/mL respectively. These data indicated that *aac(6′)-Ib-cr* acted synergistically with the *oqxABR* cluster in Tn*6010* to confer intermediate resistance (MIC ≥ 0.5 µg/mL) to ciprofloxacin. On the other hand, the chromosomal *oqxABR* cluster alone only caused a mild reduction in ciprofloxacin susceptibility, whereas *oqxAB* itself was found to have no impact on drug susceptibility even when the regulator gene *oqxR* was

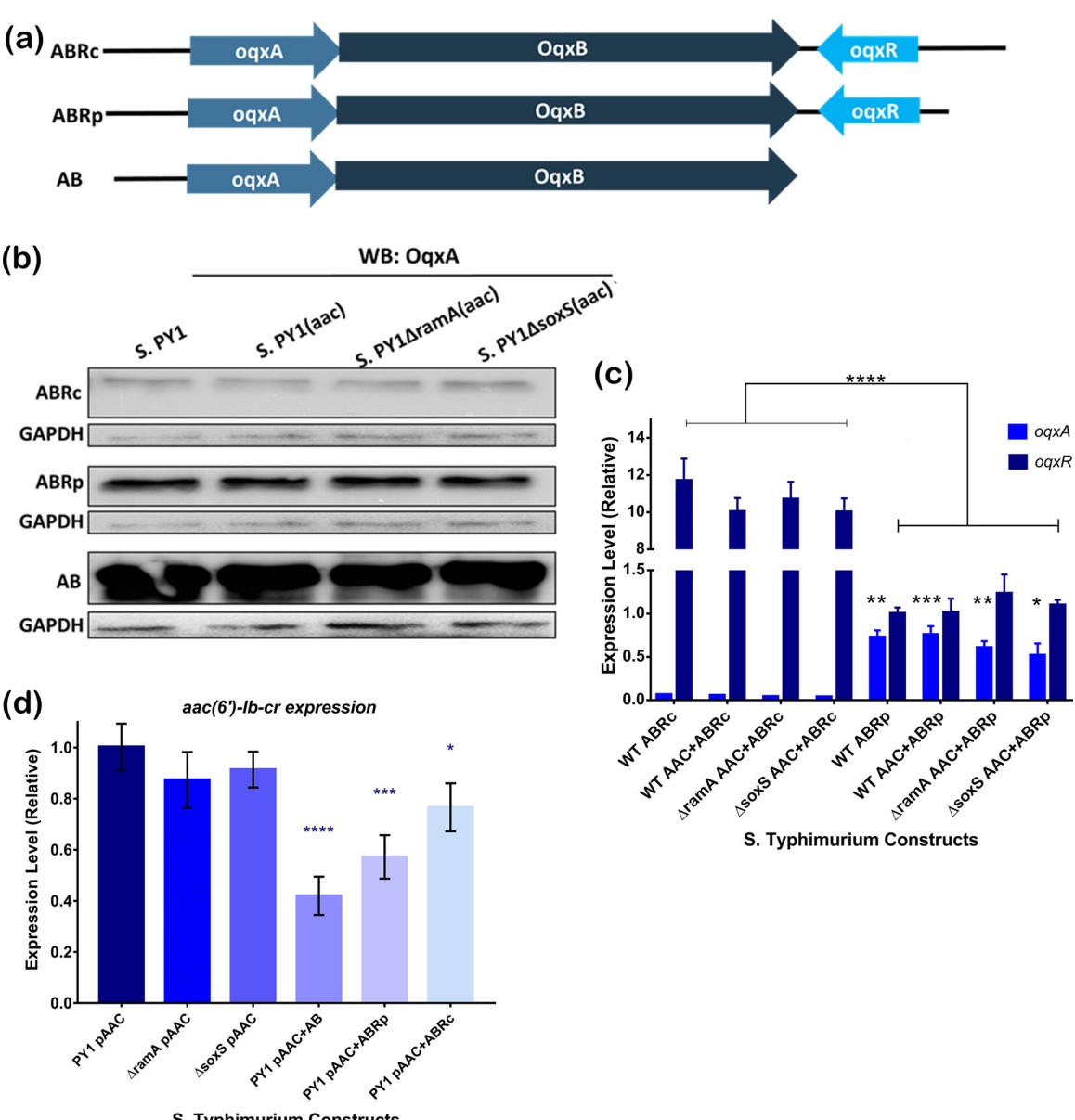

**FIG 1** Expression of *oqxAB* and *oqxR* in different genetic constructs in *S.* Typhimurium PY1. (a) Diagrammatic illustration of three constructs generated in this study. (b) Quantification of OqxAB production in *S.* Typhimurium strains carrying different constructs by Western blotting with anti-OqxA antibody. Anti-GADPH (glyceraldehyde-3-phosphate dehydrogenase) was used as a normalization control. (c) Relative levels of *oqxA* and *oqxR* expression in *S.* Typhimurium PY1 carrying different constructs. (d) Relative levels of *aac (6')-Ib-cr* expression in *S.* Typhimurium PY1 carrying different constructs. AB, constructs carrying the *oqxAB* genes but not *oqxR*; ABRc, constructs carrying the *oqxABR* locus from *K. pneumoniae* MGH78578; ABRp, constructs carrying *oqxABR* locus from *S.* Typhimurium ST06-53; PY1, *S.* Typhimurium 14028 wild-type strain; ΔramA, PY1Δ*ramA*; ΔsoxS, PY1Δ*soxS*. The data were analyzed using two-way analysis of variance (ANOVA) using Prism. *, $P < 0.01$; **, $P < 0.001$; ***, $P < 0.0001$; ****, $P < 0.00001$.

absent (Table 1). These findings prompted us to investigate why the plasmid containing the *oqxABR* cluster causes a higher degree of elevation in the MIC of CIP in the presence of *aac(6')-Ib-cr*, compared to the chromosomal *oqxABR*, yet the *oqxAB* or *oqxABR* cluster alone has little effect on CIP MIC of *S.* Typhimurium.

**Regulated expression of *oqxAB* in Tn*6010* configuration confers ability to encode resistance.** Based on the observations described above, sequence alignment of the plasmid-borne and chromosomal *oqxAB* gene showed that the upstream region of *oqxR* in these two clusters were different. We hypothesized that this discrepancy may alter the promoter activity of *oqxR* and subsequently lead to a differential expression level of this gene, which in turn causes altered repressor activity. Since the

combination of *aac(6′)-Ib-cr* and *oqxABR*p only mediated expression of intermediate resistance to CIP, we therefore used pAAC-AB, pAAC-ABRp, and pAAC-ABRc to carry out the following analyses. Western blotting using OqxA-specific antibody confirmed that only a trace amount of OqxA was produced in *S.* Typhimurium PY1 carrying pAAC-ABRc, whereas a substantial level of OqxA was detectable in organisms carrying pAAC-ABRp (Fig. 1b). When *oqxR* was not present, as in the case of pAAC-AB, the expression levels of *oqxAB* genes were found to increase dramatically (Fig. 1b). The data confirmed that *oqxR* indeed negatively regulated the expression level of *oqxAB*. We then further tested whether the discrepancy in OqxA production levels in pAAC-ABRp and pAAC-ABRc was due to differential expression levels of *oqxR*. By performing qPCR analysis, we found that the transcription product of *oqxR* in pAAC-ABRc was about 10-fold more abundant than that of pAAC-ABRp (Fig. 1c). Consistently, the transcription level of *oqxA* was found to be significantly lower in organisms carrying pAAC-ABRc than in those carrying pAAC-ABRp (Fig. 1c). Interestingly, assessment of the degree of correlation between MIC and the expression level of OqxAB suggested that moderate expression of OqxAB in pAAC-ABRp-bearing organisms conferred the highest CIP MIC, with the lower expression of OqxAB in strains carrying pAAC-ABRc mediating a slight increase in CIP MIC. Surprisingly, high-level expression of OqxAB in strains carrying the pAAC-AB construct exhibited no effect on CIP MIC. To further investigate this phenomenon, we analyzed the expression levels of *aac(6′)-Ib-cr* in different *S.* Typhimurium PY1 constructs using qPCR (Fig. 1d). Compared with that in *S.* Typhimurium strains which carried pAAC only, the expression level of *aac(6′)-Ib-cr* was reduced by 50% in strains that carried pAAC-AB, 40% in strains carrying pAAC-ABRp, and 20% in strains carrying pAAC-ABRc. The significant reduction in *aac(6′)-Ib-cr* expression in strains harboring pAAC-AB may account for their increased CIP susceptibility compared to those with pAAC-ABRp and pAAC-ABRc. In pAAC-ABRc-bearing strains, the slightly reduced expression of *aac(6′)-Ib-cr* may be compensated for by the addition of *oqxAB*; while the expression level of OqxAB was greatly suppressed by *oqxR*, which resulted in a poor reduction of CIP susceptibility. In pAAC-ABRp-bearing strains, the high expression level of OqxAB may overcome the effect of the slightly suppressed *aac(6′)-Ib-cr* expression and resulted in the lowest CIP sensitivity among strains carrying different constructs. The exact mechanism that causes reduced *aac(6′)-Ib-cr* expression is not known, but it could be due to the high fitness cost exerted by increased OqxAB production, as in the case of expression of other efflux pumps, such as AcrAB, in *S.* Typhimurium (15).

**Relatively high expression level of *oqxAB* in Tn*6010* is due to partial loss of repressive function of OqxR.** Like other local repressors, OqxR has an effect on transcriptional regulation of *oqxAB* which has been proposed to involve direct binding to the upstream region of *oqxA* (9). In this study, the promoter region of *oqxA* was first predicted by Softberry, and the transcription start site (TSS) was determined by 5′RACE. The identified TSS of *oqxA* was found to be 135 bp upstream from the start codon, which corroborated well with the predicted −35 to −10 bp promoter region. This region was subsequently amplified and incubated with purified OqxR prior to electrophoretic mobility shift assay (EMSA) analysis, which confirmed the interaction between this repressor and the promoter region (Fig. 2a). In view of the elevated *oqxR* expression in pAAC-ABRc and the observation that the *oqxA* transcription level in pAAC-ABRp also increased, it was believed that the plasmid-borne *oqxR* gene may encode a lower repressor activity due to the truncated promoter region in Tn*6010*. To test this hypothesis, TSSs of the *oqxR* gene in the plasmid and the chromosome were determined by 5′RACE using mRNA extracted from *S.* Typhimurium ST06-53 and *K. pneumoniae* MGH78578, respectively. As expected, two different TSSs of *oqxR* could be mapped from the two strains. The TSS of *oqxR* identified from the chromosome of *K. pneumoniae* was found to be 62 bp upstream from the opening reading frame of *oqxR*; a region which has been removed and replaced by IS*26* sequences in the plasmid version. The predicted −10 to −35 bp promoter region also aligned well with the identified TSS (Fig. 2b). Contrary to its chromosomal counterpart, transcription of *oqxR* started at 33 bp inside the ORF of the plasmid-borne *oqxR* gene carried by *S.* Typhimurium.

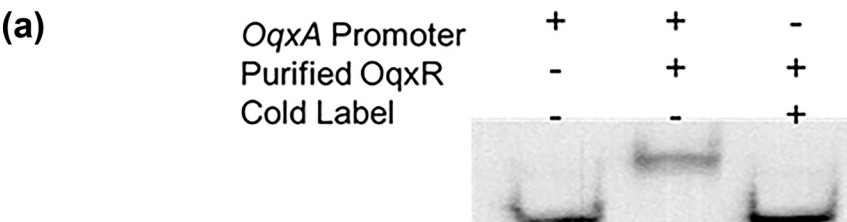

**(b)** Varied Transcription Start Site of *oqxR*

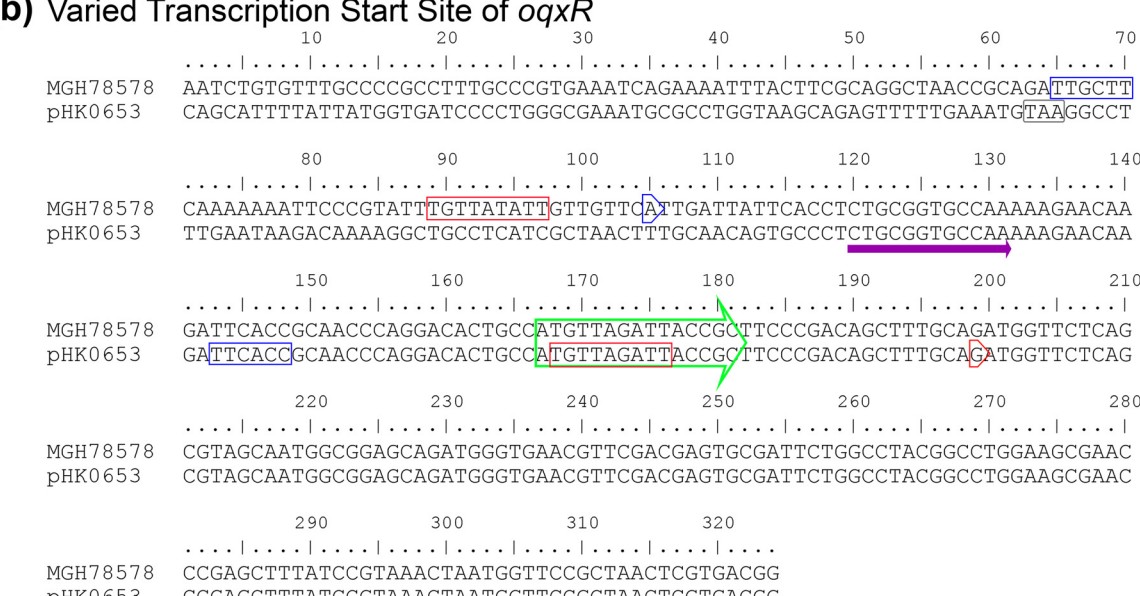

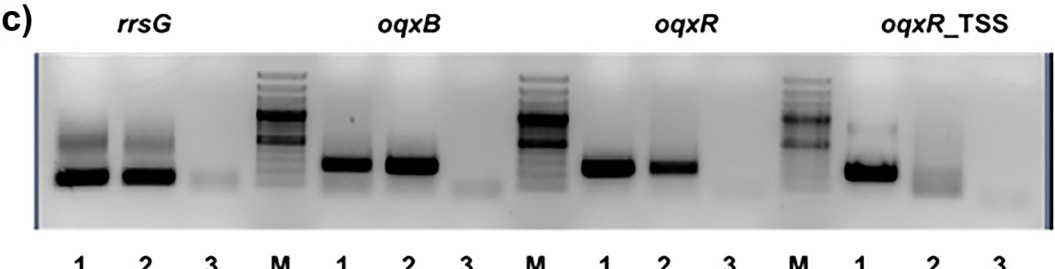

**FIG 2** Effect of variation in *oqxR* upstream sequence on expression of the *oqxR* gene. (a) Binding of OqxR on the promoter region of *oqxAB*. (b) Transcription start site (TSS) of *oqxR* in the chromosome of *K. pneumoniae* and Tn*6010*, as determined by 5′-RACE system and Sanger sequencing. The TSSs of *oqxR* from MGH78578 and pHK0653 are bracketed in the blue and red arrow boxes, respectively. Purple arrow denotes the binding sites of primers for TSS verification. The predicted *oqxR* promoter −35 and −10 motifs are bracketed in blue and red rectangle boxes, respectively. Green arrow box denotes the coding DNA sequence of *oqxR*. The gray rectangle box denotes the stop codon of IS*26*. (c) Verification of *oqxR* TSS by RT-PCR. Lane 1, *K. pneumoniae* MG78578; lane 2, *S.* Typhimurium ST06-53; lane 3, negative control.

Semi-quantitative reverse transcription PCR (RT-PCR) with primers targeting the region of *oqxR* immediately upstream of the TSS did not observe its expression in *S.* Typhimurium ST06-53, verifying the location of TSS (Fig. 2c). Taken together, our finding confirmed that the truncation of the original promoter region of *oqxR* by IS*26* in the Tn*6010*-mediated form does affect transcription of mRNA and the protein sequence, which in turn alters the repressor activity of its gene product.

**Global regulatory genes exhibit mild effect on expression of *oqxAB* in *S.* Typhimurium.** To evaluate the potential role of the global regulator genes in Tn*6010*-*oqxAB* expression, *S.* Typhimurium PY1 Δ*ramA* and Δ*soxS* knockout mutants were created, into which the pAAC-AB, pAAC-ABRc and pAAC-ABRp constructs were transformed. No significant changes in antimicrobial susceptibility were noted among *S.*

Typhimurium mutants carrying various constructs. The production of OqxAB was observed in all *S.* Typhimurium PY1 test strains, and a gene expression study showed that the *oqxAB* expression levels in all transformants were similar (Fig. 1b and c), except that a slight variation in *oqxA* expression level was observed in ΔramA and ΔsoxS knockout mutants carrying pAAC-ABRp. This finding indicates that *oqxAB* is constitutively expressed due to the loss of an effective local repressor *oqxR* and is subject to transcriptional regulation by common global regulators in *S.* Typhimurium to some extent.

## DISCUSSION

*oqxAB* and *aac(6′)-Ib-cr* are well recognized as PMQR genes which mediate reduced susceptibility to ciprofloxacin. The high prevalence of *oqxAB*-carrying *S.* Typhimurium in China, where fluoroquinolones have been extensively used in cattle and poultry farming, provided evidence of the relationship between *oqxAB* and fluoroquinolone resistance (16). To further investigate the role of *oqxAB* in fluoroquinolone resistance, *oqxAB*, *oqxABR*c, and *oqxABR*p were cloned into *S.* Typhimurium and subjected to MIC testing. Our data confirmed, for the first time, that the contributions of each of these genes to ciprofloxacin resistance are negligible. Only the combination of both genes exhibited the TMQR gene phenotype. This phenomenon aligned with a previous report that an *oqxAB*-bearing, plasmid-transformed *S.* Typhimurium may not have been phenotypically resistant to ciprofloxacin but was highly prevalent in a region with intense selection pressure from fluoroquinolone usage (17). In addition, the expression of *aac(6′)-Ib-cr* appeared to be inversely proportional to that of *oqxAB* (Fig. 1d). Interestingly, while *oqxAB* conferred reduced ciprofloxacin susceptibility, it was observed that the growth rates of pAB and pAAC+AB transformants were significantly lower than those of the transformants of other constructs. Referring to the MIC results in this study, the construct pAB appeared to have higher CIP susceptibility than the vector control (Table 1). We hypothesized that, while the *oqxAB* efflux pump may confer certain survival advantages for of *S.* Typhimurium in the presence of fluoroquinolones, it results in an extra fitness cost. It may be important for *S.* Typhimurium to control *oqxAB* expression levels in a manner which combats antimicrobials without producing too much fitness stress. Hence, the interaction between *aac(6′)-Ib-cr* and *oqxAB* gene products could affect TMQR phenotype expression and requires further investigation.

Another novel part of this study was to understand the mechanism of regulation of the plasmid-borne *oqxAB* efflux pumps. Our data are consistent with previous findings that the *oqxR* gene product is a repressor of *oqxAB* in *K. pneumoniae* (9). The result of an EMSA between the *oqxA* promoter and the purified OqxR protein showed the molecular interaction between them. The function of OqxR can be demonstrated by the suppressed OqxAB production upon integration with *oqxR* in the pABRc construct. As expected, reduction of *oqxR* expression together with increased *oqxAB* expression in pABRp was observed, providing evidence that *oqxR*, being encoded on a plasmid, is deficiently expressed. The GntR type transcriptional regulator OqxR shares common structures with other members in the family. It contains a typical helix-turn-helix (HTH) domain and a ligand-binding domain (18). Thus, it is believed that OqxR exhibits its repressive effects in a fashion similar to that of other transcription factors such as AcrR, in which its HTH domain binds to the inverted repeat (IR) nucleotide sequences located within the promoter region of *oqxAB* in dimer form to avoid transcription by RNA polymerase. This strongly supports the idea that the decreased expression of *oqxR* in the plasmid-encoded structure was due to the truncation of its upstream region, leading to changes in nucleotide sequences which resulted in the loss of or damage to promoter-binding sites. As the result, reduction in *oqxR* expression leading to increase in *oqxAB* expression.

In addition to local regulation, expression of OqxAB is also shown to be affected by global regulators such as *ramA* and *soxS*. Our data showed the absence of *ramA* and *soxS* resulted in slightly reduced *oqxA* expression in the ΔramA and ΔsoxS mutants

(Fig. 1c). While the level of this is minimal and it exhibits no effect on expression of TMQR phenotypes, it has been reported that in *K. pneumoniae*, where endogenous *oqxAB* resides, loss of *ramR*, resulting in overproduction of RamA, would lead to the overexpression of *oqxAB* (10). Based on the partial sequences of plasmids deposited into GenBank, the TMQR determinants in *oqxABR* are all found to be mediated by the transposon Tn*6010*, which was first sequenced along with an *E. coli* IncX plasmid carrying *oqxAB* in 2008 (5). It is believed that the IS*26* transposition event captured the whole *oqxABR* locus from *K. pneumoniae*, but not the *rarA* gene located upstream of *oqxA*. A previous study has demonstrated that *rarA*, a homologue of *ramA*, may exhibit an induction effect on *oqxAB* expression and subsequently confer reduced antimicrobial susceptibility in *K. pneumoniae* (9). Considering the fact that the *rarA* product binds to the promoter region of *oqxAB*, and also its genetic homology with *ramA*, we surmise that in *Salmonella* Typhimurium, a functional defect due to a lack of the *rarA* gene can be compensated for by the effect of the product of the homogenous *ramA* gene. Based on the results of the gene expression study, reduced *oqxAB* expression was observed in strains upon the removal of *ramA* and *soxS*, suggesting that intrinsic transcriptional factors, which have distal regulatory effects, regulate not only endogenous genes, but also those acquired through plasmid uptake or transposition. Interestingly, the fact that the absence of *ramA* in *E. coli* does not hinder the substrate extrusion capacity of OqxAB, as illustrated elsewhere (2, 19), implies that this TMQR may be controlled by different regulatory mechanisms in this bacterial species.

Nevertheless, we only used *S.* Typhimurium as our model strain in this study to test for the molecular regulation of the *oqxABR* operon and the effects of *ramA* and *soxS*. This limited the study, which can only confirm the results in *S.* Typhimurium and particular mutants. In future works, we will try to expand our study to various serovars and different bacterial species in order to get a more complete picture for the regulation of *oqxABR* operon and other TMQR genes.

## CONCLUSION

Our work revealed the constitutively expressed nature of the plasmid-encoded TMQR elements *aac(6′)-Ib-cr* and *oqxAB*. The plasmid-borne *oqxAB* is mainly regulated by the local repressor *oqxR*. The truncated *oqxR* sequence in plasmid-borne, Tn*6010*-mediated *oqxABR* resulted in supressed expression of OqxR, which deficiently supressed the expression of OqxAB and subsequently resulted in a ciprofloxacin-resistant phenotype in *S.* Typhimurium. In addition, our findings showed that only optimal expression of the TMQR genes *aac(6′)-Ib-cr* and *oqxAB* may confer fluoroquinolone resistance in *S.* Typhimurium. Global transcriptional regulators in *S.* Typhimurium may be crucial in plasmid-borne *oqxAB* regulation and help overcome the effect of the local repressor OqxR, demonstrating that global regulatory mechanisms are capable of controlling the expression of plasmid-borne genes. Further studies are warranted to elucidate the actual events underlying the interaction between the products of *aac(6′)-Ib-cr*, *oqxAB*, *oqxR*, and *ramA* in *Salmonella* spp., as well as the mechanisms regulating the expression of TMQR genes in other members of *Enterobacteriaceae*.

## MATERIALS AND METHODS

**Bacteria strains and vectors.** Bacterial strains used in this study are listed in Table S1 in the supplemental material. *S.* Typhimurium strain 14028 (PY1, SGSC2262) was obtained from our laboratory collection. Propagation of bacteria was conducted in LB medium with/without antibiotics at 37°C. The cloning vector used was the pACYCDuet-1 backbone (Addgene no. 71147) which contained the ampicillin resistance gene for selection purposes.

**Genetic analysis for *oqxABR* locus.** The complete genomic sequences of *K. pneumoniae* MGH75878 (GenBank ID: CP000647) and the plasmid sequence of pHK06-53 from *S.* Typhimurium (GenBank ID: KT334335) were retrieved from NCBI. The *oqxABR* locus was extracted from both and compared using ClustalW alignment. The IS element was identified and located using Isfinder (20).

**Generation of *oqxABR* operon constructs.** Three constructed plasmids, namely, pABRp, pABRc, and pAB, which represented the plasmid-borne *oqxAB* operon; the chromosomal *oqxAB* operon, and *oqxAB* alone, respectively, were generated by using the modified pACYCDuet-1 as a cloning vector. The modified pACYCDuet-1 vector (pACYCDuet-AmpR) was constructed using pACYCDuet-1 (Addgene no. 71147)

with a replacement of *cmR* by *AmpR* from pET-15b plasmid (Addgene no. 69661-3). Plasmid modification was done using a Gibson Assembly cloning kit (E2611; New England BioLabs). The pACYCDuet-1 provided two separate cloning sites for the simultaneous insertion of two gene fragments. DNA fragments of the test genes were amplified by Primestar GXL polymerase and purified by gel electrophoresis. The vector and inserted genes were double-digested with restriction enzymes and ligated by T4 ligase. pABRc was constructed by cloning the *oqxABR* operon and the 353-bp region upstream from *oqxR* in *K. pneumoniae* MGH78578 (GenBank ID: CP000647). pABRp was constructed by cloning the *oqxABR* locus and the 84-bp region upstream from *oqxR*, resembling the configuration in Tn*6010* from pHK06-53 (GenBank ID: KT334335). *AB* was constructed by cloning the *oqxAB* region and the 315-bp sequence upstream from *oqxA* in Tn*6010* from pHK06-53. In addition, the *aac(6')-Ib-cr* sequence originating from pHK06-53 was inserted into pACYCDuet-AmpR, as well as the pABRp, pABRc, and pAB constructs, to produce four constructs: pAAC, pAAC-ABRp, pAAC-pABRc, and pAAC-AB. The cloning procedures were the same as those described above. All primers used are listed in Table S2.

**Antimicrobial susceptibility tests.** The MICs of seven antimicrobials (ciprofloxacin, gentamicin, nalidixic acid, olaquindox, cefotaxime, and chloramphenicol) were determined for all test strains and interpreted by the broth microdilution method according to CLSI guidelines (21). Cation-adjusted Mueller-Hinton broth was used as the culture medium. *Escherichia coli* ATCC 25922 was used as quality control. The test was repeated three times.

**Western blotting of OqxA.** The constructs of *Salmonella* Typhimurium were first streaked on LB agar plate to ensure no contamination of the stock. The single colonies from each construct were inoculated in LB broth and incubated overnight at 37°C with shaking. After that, the broth cultures of each construct were reinoculated in fresh LB broth and incubated at 37°C with shaking. Each construct underwent 3 repeated setups to ensure consistent results. Each test strain was grown in LB medium until the absorbance at 600 nm reached 0.5. One mL of the culture was centrifuged and resuspended in 400 $\mu$L SDS loading buffer, then boiled for 10 min. Solubilised proteins were separated by SDS-PAGE and subsequently transferred to a polyvinylidene difluoride membrane through a semi-dry transfer apparatus. Western blotting was carried out by probing the membrane with rabbit anti-OqxA monoclonal antibody, followed by goat anti-rabbit IgG. The signal was visualized by the addition of horseradish peroxidase substrate. *Salmonella* glyceraldehyde-3-phosphate dehydrogenase-specific antibody was used as an endogenous loading control.

**RNA extraction and qRT-PCR.** From the same batch of bacterial culture prepared for the Western blotting of OqxA, total RNA was extracted using a Qiagen RNeasy Protect Bacteria Mini Kit, followed by DNase treatment. The quality and quantity of RNA were determined using a Nanodrop spectrophotometer. One $\mu$g of total RNA was subjected to reverse transcription using SuperScript III Reverse Transcriptase from Life Technologies. qRT-PCR mixture was prepared using Life Technologies SYBR Select Master Mix. PCR was performed using an Applied Biosystems Quant Studio 3 system. The primers used in qPCR are listed in Table S2. A melt curve analysis of the PCR product was performed to ensure detection specificity. Expression levels of the test genes were normalized with that of the housekeeping gene, which encodes DNA gyrase subunit B. Gene expression level data were processed using GraphPad Prism 7.0, and statistical significance was calculated by analysis of variance (ANOVA).

**Determination of transcription start site of *oqxR*.** The TSS of the *oqxR* gene in *K. pneumoniae* and *S.* Typhimurium ST06-53 was determined by 5' rapid amplification of cDNA end (5'RACE), using an Invitrogen 5'RACE kit. Briefly, RNA was extracted from both strains, and contaminating DNA was removed by DNase treatment, followed by conversion to cDNA by the use of the primer GSP1. After SNAP purification and TdT tailing, the dc-tailed cDNA was amplified using the abridged anchor primer, GSP2, and GSP3. The purified PCR product was TA-cloned into the pCR2.1-TOPO vector (Invitrogen K452001) and subsequently transformed into DH5$\alpha$-T1. Cloning procedures followed the instructions provided by the manufacturer. Positive-insert transformants were selected by ampicillin and blue-white screening. The cloned sequence was confirmed by Sanger sequencing. The junction between the C tail and the start site of the *oqxR* open reading frame was regarded as the transcriptional start site. The putative −10 and −35 hexamers were predicted by performing SoftBerry analysis of the intergenic region (22).

**Generation of *ramA* and *soxS* knockout mutants.** *S.* Typhimurium PY1 knockout mutants were generated by the pKD46 homologous recombination system as previously described (23). Briefly, helper plasmid pKD46 was electroporated into *S.* Typhimurium PY1-competent cells. Expression of $\lambda$-recombinase was induced by addition of L-arabinose. The PCR product of the kanamycin resistance gene, flanked by the FRT-sequence and the 50-bp homologous sequence, was cloned into pKD4 plasmid, probed by the primers listed in Table S2, and subsequently electroporated into recombinase-induced cells. Mutants were selected on a LB agar plate supplemented with 50 $\mu$g/mL kanamycin. The identity of knockout mutants was verified by PCR using primers listed in Table S2.

**Electrophoretic mobility shift assay.** The open reading frame of *oqxR* was cloned into a pET28 expression vector, followed by induction of expression in *E. coli* strain BL21. The expressed protein was purified by using a nickel-chelated NTA column and performing size exclusion chromatography in an ÄKTA protein purification system. The promoter region of *oqxA* was amplified from *K. pneumoniae* MGH78578 by PCR using primers listed in Table S2. EMSA was performed using a Gel Shift Kit, 2nd generation (Roche). Briefly, amplified DNA was labeled with digoxigenin and incubated with OqxR at room temperature for 30 min, followed by electrophoresis in a precast 6% acrylamide non-denaturing gel with ice-cold Tris-borate-EDTA buffer. The contents were transferred to a nylon membrane by a semi-dry electrotransfer system, followed by UV cross-linking. The membrane was probed against anti-dig antibody prior to addition of the substrate for chemiluminescence detection.

## SUPPLEMENTAL MATERIAL

Supplemental material is available online only.

**SUPPLEMENTAL FILE 1**, PDF file, 0.6 MB.

## ACKNOWLEDGMENT

This work was supported by the Basic Research Fund of Shen-zhen (20170410160041091) and a CityU internal grant (9380110).

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
