## [Reviewer comments · Microbiology Spectrum]

Microbiology Spectrum

Transcriptional regulation and functional characterization of the plasmid-borne *oqxAB* genes in *Salmonella* Typhimurium

Bill Kwan-wai CHAN, Marcus Ho-yin WONG, Edward Wai-Chi Chan, and Sheng Chen

Corresponding Author(s): Sheng Chen, CityU of Hong Kong

Review Timeline:

Submission Date:	November 6, 2021
Editorial Decision:	January 27, 2022
Revision Received:	February 22, 2022
Accepted:	March 5, 2022

Editor: Sandeep Tamber

Reviewer(s): Disclosure of reviewer identity is with reference to reviewer comments included in decision letter(s). The following individuals involved in review of your submission have agreed to reveal their identity: Hongning Wang (Reviewer #1)

Transaction Report:

DOI: <https://doi.org/10.1128/spectrum.02170-21>

January 27, 2022

Dr. Sheng Chen
CityU of Hong Kong
Department of Infectious Diseases and Public Health
Department of PH
Kowloon
Hong Kong

Re: Spectrum02170-21 (Transcriptional regulation and functional characterization of the plasmid-borne oqxAB genes in *Salmonella Typhimurium*)

Dear Dr. Sheng Chen:

Link Not Available

Sincerely,

Sandeep Tamber

Journals Department
Editor comments:

As indicated by both reviewers, please expand on the background information provided in the Introduction and ensure the methods are described fully so they can be reproduced by others. The manuscript requires further editing and revising to improve clarity. The link below provides information on language editing services for your consideration. If these items in addition to the points below are properly addressed, I will consider the publication of this manuscript.

<https://journals.asm.org/language-editing-services>

Reviewer comments:

Reviewer #1 (Comments for the Author):

The authors determined the transcriptional regulation and functional characterization of the plasmid-borne oqxAB genes in *Salmonella Typhimurium*. It has some interest for readers. I have some issues.

(1) In INTRODUCTION, oqxAB gene was found more than ten years ago. I suggest that the authors cite some reviews in the introduction (Drug Resist Updat. 2016 Nov;29:13-29. Clin Microbiol Rev

. 2019 Aug 14;32(4):e00007-19.)

(2) Line 85, *S. Typhimurium* strain 14028s, is it a ATCC strain?

(3) Line 87, where the pACYCDuet-1 comes from?

(4) Lines 178-182, please list the reference.

(5) Lines 184-185, The ciprofloxacin (CIP) MIC of the transformant was 2~fold lower than the vector control? lower or higher?

Reviewer #2 (Comments for the Author):

Review for the manuscript titled: "Transcriptional regulation and functional characterization of the plasmid-borne *oqxAB* genes in *Salmonella Typhimurium*" by Bill Kwan-wai CHAN et al. submitted for publication in *Microbiology Spectrum*

In this study aim to evaluate the actual role of the plasmid-borne *oqxAB* genes in mediating changes in fluoroquinolone susceptibility and to elucidate the mechanism by which expression of this efflux pump is regulated in a strain of *S. Typhimurium*. The authors are taking multistep approach that includes the creation of various constructions of the strain and comparing the expression of genes and the resistance phenotype.

The authors are also estimating the effect on the MIC of CIP in the presence of the *aac(6')-Ib-cr* gene. Even though the effect of the latter on resistance to quinolone was previously demonstrated in multiple studies to be small or even absent and that the overall effect found to *oqxAB* genes with/out the *aac(6')-Ib-cr* seems to be relatively small, studies as this are essential for confirming in vitro mechanisms of resistance.

There are few limitations in the current study that should be addressed to allow better understanding of the work flow and rationale, and also the limitations of such study that was conducted using a single lab strain of a specific *Salmonella* serotype. In the current version, I am not completely sure how many repetitions were done for each experiment and under which conditions - this information should be stated clearly in the material and methods section.

Some general comments:

- The abstract section is not well written - study objective is different than the one presented in the manuscript and there is no clear explanation of the problem that has initiated this study
- The introduction section is lacking important information such as what is regarded as "reduced susceptibility" this problem is repeated in the study. Also, additional citations from the literature are required
- The material and methods are too general and does not detail the multistep approach taken here. Also, there is no information on statistical analysis that was done and should be included here. The materials section should include a simple flow chart that demonstrate the steps taken in the analysis. In the current format it is hard to follow the steps taken here. Also see the important comment above regarding the number of repetitions.
- The results section should be separated from the discussion. The results need to be presented according to the steps detailed in the material and methods section. Including the main outcomes from the tables and figures
- The discussion section should be written separately. Additional work on the literature review and adding citations is needed to support the explanation of the results. As part of this section the authors should address the limitations of such study.

More specific comments:

Line 24 - "Co-" should not be in bold

Lines 26-29 - it is not clear from what is written what is the difference in the phenotype between *Salmonella* and *Klebsiella* and how is this related to *aac(6')-Ib-cr* gene mentioned before

Line 30 - "*oqxAB*" should be in italics

Line 34 - indicate the MIC cutoff used to define reduced susceptibility

Line 46 - foreign bacterial host is referring to *Salmonella*?

Line 57 - remove "which"

Line 62-63 - in a previous study by the authors ("*PMQR* genes *oqxAB* and *aac(6_)Ibcr* accelerate the development of fluoroquinolone resistance in *Salmonella typhimurium*") the presence of a mutation on *gyrA* gene was also described (this was also described in "*Spread of oqxAB in Salmonella enterica serotype Typhimurium predominantly by IncHI2 plasmids*"). Is the presence of *oqxAB* and *aac6-ib-cr* can result in CIP resistance without the *gyrA* mutation?

Line 64 - please provide the MIC value or define clearly what is reduced susceptibility to ciprofloxacin

Lines 73-75 "While global regulators in *S. Typhimurium*, such as *ramA*, *marA* and *soxS*, play an integral role in regulating the host's endogenous efflux gene *acrAB*" please provide citations to support this statement

Lines 77-78 - I am not completely clear about the reasoning behind this hypothesis. Please clarify

In line 30 "Objective: to investigate the regulation of the plasmid-borne *oqxAB* in *Salmonella*" than on lines 80-82 "The aim of this study is to evaluate the actual role of the plasmid-borne *oqxAB* genes in mediating changes in fluoroquinolone susceptibility and elucidate the mechanism by which expression of this efflux pump is regulated in *S. Typhimurium*" - please be consistent

Line 85 - what is known about the genome of this strain - specifically on mutations (synonymous and non-synonymous) in *gyrA*, *gyrB*, *parC* and *parE* genes and on possible mutations in *ramR*, *marR*, *soxR* and *acrR* genes

Also, what is the resistance phenotype (including MIC values) for CIP and NAL?

Lines 107-108 - what about constructs without *aac*?

Line 116 - How were the repeated test results treated - as an average?

Line 138 - what about *marA* knockout?

Line 175 - please correct "dicussion" to "discussion"

Also, why are the results mixed with the discussion? it would be better to present separately

Lines 178 -180 - which previous studies? - citations are required. Also, MIC in E. coli does not necessarily indicate what will be the MIC in Salmonella

Line 184 - do you mean ~2 fold? Please correct

Line 185 - significantly - were there any statistical tests used here?

Line 186-190 this information should be part of the material and methods

Lines 190-193 was comparing the chromosomal and plasmid Oqx complex was part of the aims of this study?

Line 197 - 4 fold higher and in the same level? Please clarify

Line 199 - "Salmonella" - even though generalizing the findings to Salmonella may be tempting, here only a single serovar and even more a single strain was tested. Please check "Embracing Diversity: Differences in Virulence Mechanisms, Disease Severity, and Host Adaptations Contribute to the Success of Nontyphoidal Salmonella as a Foodborne Pathogen" by Rachel A. Cheng et al. 2019 on the limitations of such approach

Lines 204 -205 which field strains are you referring to here? Please provide details on the host, collection time, location, etc. Also, in 205 correct to ~0.25-1

Line 206 - the contribution of aac to CIP resistance was previously demonstrated in many other studies to be minimal if at all exist. Please cite!

Line 208 - such information should be included in the methods. In general, the methods section should provide a step by step description of the analysis done. This is not done here. Instead steps and some justification are only presented as part of the results

Line 211 - "intermediate" - please clarify the term (see my comment regarding "reduced susceptibility")

Lines 215-217 - These conclusions which are based on a single strain of certain Salmonella serovar for which the plasmids were introduced once and MIC was estimated three times - this is highly problematic to deduce from this experiment on the contribution of AAC or ABRp to CIP resistance

Line 219 - again based on a single trial on a single clone

Line 221 - significantly - was there any statistical test conducted?

Line 254-255 - how is the pAAC-ABRc results are explained?

Line 290 - what about mar gene

Line 300 - "PMGR" ? - do you mean PMQR

Line 302 - the small/absent effect of aac gene, at least, is well documented in many other studies - including those testing a wide variety of Salmonella serotypes and using WGS in comparison to the phenotype

Lines 302-303 - also the additive contribution - mainly in addition to chromosomal mutations was well documented

Figure 1 - which statistical tests and which comparisons should be stated clearly in the methods and not only here

Line 376 - relative - what was the reference? pAAC?

Figure 2 - please clarify the split in TSS - for example - what is the meaning and how it was determined that the base in location 199 is part of the TSS of pHK0653 and not for example a base in location 196? or 180?

Staff Comments:

Preparing Revision Guidelines

Please return the manuscript within 60 days; if you cannot complete the modification within this time period, please contact me. If you do not wish to modify the manuscript and prefer to submit it to another journal, please notify me of your decision immediately so that the manuscript may be formally withdrawn from consideration by Microbiology Spectrum.

Responses to reviewers' and Editor's comments

Editor comments:

As indicated by both reviewers, please expand on the background information provided in the Introduction and ensure the methods are described fully so they can be reproduced by others. The manuscript requires further editing and revising to improve clarity. The link below provides information on language editing services for your consideration. If these items in addition to the points below are properly addressed, I will consider the publication of this manuscript.

<https://journals.asm.org/language-editing-services>

Response: All comments have been addressed.

Reviewer comments:

Reviewer #1 (Comments for the Author):

The authors determined the transcriptional regulation and functional characterization of the plasmid-borne *oqxAB* genes in *Salmonella* Typhimurium. It has some interest for readers. I have some issues.

(1) In INTRODUCTION, *oqxAB* gene was found more than ten years ago. I suggest that the authors cite some reviews in the introduction (Drug Resist Updat. 2016 Nov;29:13-29. Clin Microbiol Rev. 2019 Aug 14;32(4):e00007-19.)

Response: Thank you for your suggestion. The mentioned article has been cited and the term PMQR has been revised to TMQR as well.

(2) Line 85, *S.* Typhimurium strain 14028s, is it a ATCC strain?

Response: This strain was requested from Salmonella Genetic Stock Centre (SGSC), code SGSC 2262, which was equivalent to ATCC14028s.

(3) Line 87, where the pACYCDuet-1 comes from?

Response: The original plasmid was ordered from Merck Millipore company, product code 71147. It was modified by ourselves for this project.

(4) Lines 178-182, please list the reference.

Response: The reference has been added.

(5) Lines 184-185, The ciprofloxacin (CIP) MIC of the transformant was 2~fold lower than the vector control? lower or higher?

Response: According to the result shown in Table 1, the PY1 transformed with construct AB has ciprofloxacin MIC of 2-fold lower than the vector control. We had repeated the test for 3 times which produced the same result. The lowered MIC may be due to the extremely high expression of oqxAB which conferred fitness cost to the bacterial cell.

Reviewer #2 (Comments for the Author):

Review for the manuscript titled: "Transcriptional regulation and functional characterization of the plasmid-borne oqxAB genes in Salmonella Typhimurium" by Bill Kwan-wai CHAN et al. submitted for publication in Microbiology Spectrum
In this study aim to evaluate the actual role of the plasmid-borne oqxAB genes in mediating changes in fluoroquinolone susceptibility and to elucidate the mechanism by which expression of this efflux pump is regulated in a strain of S. Typhimurium. The authors are taking multistep approach that includes the creation of various constructions of the strain and comparing the expression of genes and the resistance phenotype.

The authors are also estimating the effect on the MIC of CIP in the presence of the aac(6')-Ib-cr gene. Even though the effect of the latter on resistance to quinolone was previously demonstrated in multiple studies to be small or even absent and that the overall effect found to oqxAB genes with/out the aac(6')-Ib-cr seems to be relatively small, studies as this are essential for confirming in vitro mechanisms of resistance. There are few limitations in the current study that should be addressed to allow better understanding of the work flow and rationale, and also the limitations of such study that was conducted using a single lab strain of a specific Salmonella serotype. In the current version, I am not completely sure how many repetitions were done for each experiment and under which conditions - this information should be stated clearly in the material and methods section.

Response: Thank you for your comment. The sample preparation has been revised in methodology section.

Some general comments:

- The abstract section is not well written - study objective is different than the one presented in the manuscript and there is no clear explanation of the problem that has

initiated this study

Response: The abstract section has been revised.

- The introduction section is lacking important information such as what is regarded as "reduced susceptibility" this problem is repeated in the study. Also, additional citations from the literature are required

Response: The term reduced susceptibility has been clarified/defined. Additional citations have been added.

- The material and methods are too general and does not detail the multistep approach taken here. Also, there is no information on statistical analysis that was done and should be included here. The materials section should include a simple flow chart that demonstrate the steps taken in the analysis. In the current format it is hard to follow the steps taken here. Also see the important comment above regarding the number of repetitions.

Response: Thank you for your comment. The number of repetitions and the methods for statistical analysis have been added in the methodology section.

- The results section should be separated from the discussion. The results need to be presented according to the steps detailed in the material and methods section. Including the main outcomes from the tables and figures

Response: Thank you for your suggestions. We have separated the result from the discussion part. The result is followed with the methodology part.

- The discussion section should be written separately. Additional work on the literature review and adding citations is needed to support the explanation of the results. As part of this section the authors should address the limitations of such study.

Response: Thank you for your suggestions. We have separated the result from the discussion part. Extra information and citation has been added to discussion part.

More specific comments:

Line 24 - "Co-" should not be in bold

Response: The font has been corrected in text.

Lines 26-29 - it is not clear from what is written what is the difference in the phenotype between Salmonella and Klebsiella and how is this related to aac(6')-Ib-cr gene mentioned before

Response: This sentence has been deleted from abstract since it is stated in introduction section.

Line 30 - "oqxAB" should be in italics

Response: The word format has been changed in the text.

Line 34 - indicate the MIC cutoff used to define reduced susceptibility

Response: The MIC cut-off has been added. The number is referenced from CLSI guideline M100, 31st edition and added to the reference list.

Line 46 - foreign bacterial host is referring to Salmonella?

Response: It has been corrected.

Line 57 - remove "which"

Response: the word "which" has been removed from the text.

Line 62-63 - in a previous study by the authors ("PMQR genes oqxAB and aac(6_)Ibcr accelerate the development of fluoroquinolone resistance in Salmonella typhimurium") the presence of a mutation on gyrA gene was also described (this was also described in "Spread of oqxAB in Salmonella enterica serotype Typhimurium predominantly by IncHI2 plasmids"). Is the presence of oqxAB and aac6-ib-cr can result in CIP resistance without the gyrA mutation?

Response: Thank you for your question. From the data described in "Spread of oqxAB in Salmonella enterica serotype Typhimurium predominantly by IncHI2 plasmids", although most of the CIP resistant strains were found to carry gyrA mutation, a few of them carried WT gyrA and oqxAB and aac(6')-Ib-cr positive, e.g. GDS45, GDS147-149, which had MIC value of 0.5 to 1 µg/mL. Although the resistant is not as high as those

with *gyrA* mutation, it was a good evidence that *oqxAB* and *aac(6')-Ib-cr* could confer CIP resistant to *Salmonella* spp. even without *gyrA* mutation in the host.

Line 64 - please provide the MIC value or define clearly what is reduced susceptibility to ciprofloxacin

Response: The MIC value has been added and provided with reference.

Lines 73-75 "While global regulators in *S. Typhimurium*, such as *ramA*, *marA* and *soxS*, play an integral role in regulating the host's endogenous efflux gene *acrAB*" please provide citations to support this statement

Response: Reference has been added.

Lies 77-78 - I am not completely clear about the reasoning behind this hypothesis. Please clarify

*Response: Thank you for your question. According to the transcription start site analysis in this study, we found that the transcription product of *oqxR* from Tn6010 was truncated by 62 bp when compared to that of *oqxR* from chromosome of *K. pneumoniae*. We cloned both *oqxR* with *oqxAB* and transformed into *Salmonella* PY1 and checked for the expression level of *oqxR* and *oqxAB* by qPCR. It was discovered that the mRNA level of *oqxR* from Tn6010 was lower than that from chromosome of *K. pneumoniae*. The mRNA level of *oqxAB* was also highly suppressed by *oqxR* from chromosome of *K. pneumoniae* while the suppression reduced when *oqxR* was replaced with Tn6010 counterpart. The western blot result also well aligned with the qPCR result. So we hypothesized that the truncated *oqxR* in Tn6010 may reduce the production of *oqxR* transcript leading to weaker suppression of the expression of *oqxAB*.*

In line 30 "Objective: to investigate the regulation of the plasmid-borne *oqxAB* in *Salmonella*" than on lines 80-82 "The aim of this study is to evaluate the actual role of the plasmid-borne *oqxAB* genes in mediating changes in fluoroquinolone susceptibility and elucidate the mechanism by which expression of this efflux pump is regulated in *S. Typhimurium*" - please be consistent

Response: The objective of has been changed

Line 85 - what is known about the genome of this strain - specifically on mutations

(synonymous and non-synonymous) in gyrA,gyrB, parC and parE genes and on possible mutations in ramR, marR, soxR and acrR genes
Also, what is the resistance phenotype (including MIC values) for CIP and NAL?

Response: The MIC values for CIP and NAL are 0.0156µg/mL and 4µg/mL respectively, which is the same as vector control stated in Table 1. The genome sequence could not be found from GenBank. From the MIC, this strain should be very susceptible to fluoroquinolones and there should not have mutations on gyrA or parC genes.

Lines 107-108 - what about constructs without aac?

Response: The constructs without aac were named as pABRp, pABRc and pAB. In total we had made 7 types of constructs, including pABRp, pABRc, pAB, pAAC, pAAC-ABRp, pAAC-ABRc and pAAC-AB.

Line 116 - How were the repeated test results treated - as an average?

Response: Usually the MIC value in the three repeated test were the same. In a few cases we may got results with 1-fold deviation. We chose the 2 identical values in the 3 repeated test as the final result.

Line 138 - what about marA knockout?

Response: The marA gene was not characterized in this manuscript.

Line 175 - please correct "dicussion" to "discussion"

Response: The typing mistake has corrected in the text.

Also, why are the results mixed with the discussion? it would be better to present separately

Response: We have separated these two sections.

Lines 178 -180 - which previous studies? - citations are required.

Response: Reference has been added.

Also, MIC in E. coli does not necessarily indicate what will be the MIC in Salmonella

Response: Apart from Salmonella, oqxAB is also prevalent among E. coli. We want to stated that the presence of oqxAB in the host may not effectively induce the resistance to CIP.

Line 184 - do you mean ~2 fold? Please correct

Response: It was typing mistake of 2-fold. Has been corrected in the text.

Line 185 - significantly - were there any statistical tests used here?

Response: we have deleted it.

Line 186-190 this information should be part of the material and methods

Response: Thanks for your comments. We believe that this part is the data analysis and is suitable to be part of the results. Also, keeping this analysis will make readers better understand the data following this part. If the reviewer feels not appropriate to keep it here, we can move it to method section.

Lines 190-193 was comparing the chromosomal and plasmid Oqx complex was part of the aims of this study?

Response: Thank you for your question. Yes. Since it was found that the transcription start site of oqxR in chromosomal of Klebsiella was different from that in Tn6010, part of the aim of this study was to study the difference of them on regulation of oqxAB.

Line 197 - 4 fold higher and in the same level? Please clarify

Response: 4-fold higher was referring to the MIC level of pABRp construct, and same level was referring to the MIC level of pABRc construct. The sentence has been rephrased.

Line 199 - "Salmonella" - even though generalizing the findings to Salmonella may be tempting, here only a single serovar and even more a single strain was tested. Please check "Embracing Diversity: Differences in Virulence Mechanisms, Disease Severity,

and Host Adaptations Contribute to the Success of Nontyphoidal Salmonella as a Foodborne Pathogen" by Rachel A. Cheng et al. 2019 on the limitations of such approach

Response: Salmonella is replaced by Salmonella Typhimurium as the test was done using this Serova. We agreed that different serotypes of Salmonella exhibited different virulence and host adaption, which should be attributed to different genetic factors carried by different serotypes of Salmonella. In this study, we used one strain from a serotype as host to study the interactive contribution of two AMR genes to the fluoroquinolone resistance phenotype, which should be the correct way in consideration the diversity of different serotypes.

Lines 204 -205 which field strains are you referring to here? Please provide details on the host, collection time, location, etc.

Response: The table of the information of field strains has been added to the supplementary document.

Also, in 205 correct to ~0.25-1

Response: The text has been corrected.

Line 206 - the contribution of aac to CIP resistance was previously demonstrated in many other studies to be minimal if at all exist. Please cite!

Response: The citation of previous studies for contribution of aac to CIP resistance has been added.

Line 208 - such information should be included in the methods. In general, the methods section should provide a step by step description of the analysis done. This is not done here. Instead steps and some justification are only presented as part of the results

Response: This sentence has been deleted since we have this part in method already.

Line 211 - "intermediate" - please clarify the term (see my comment regarding "reduced susceptibility")

Response: The clarification has been added in the text.

Lines 215-217 - These conclusions which are based on a single strain of certain Salmonella serovar for which the plasmids were introduced once and MIC was estimated three times - this is highly problematic to deduce from this experiment on the contribution of AAC or ABRp to CIP resistance

Line 219 - again based on a single trial on a single clone

Response: We understand your concern. Since S. Typhimurium is one of the main pathogen causing salmonellosis worldwide, that is the reason for us to choose it as model strain for our study. I agree that it is better to run the tests on multiple serovars, but it will consume a lot of time and man-power. For this, I will mention it in the discussion part as a limitation for our study.

Line 221 - significantly - was there any statistical test conducted?

Response: We have deleted it.

Line 254-255 - how is the pAAC-ABRc results are explained?

Response: The explanation for pAAC-ABRc result has been added

Line 290 - what about mar gene

Response: We did not try with mar regulator in this study. Although we know that mar is the sole regulator for various antibiotic mechanisms, it more likely to regulate stress response under treatment of antibiotics. As both soxS and ramA were reported as efflux pump regulators in klebsiella and E. coli, we mainly conduct tests of them.

Line 300 - "PMGR" ? - do you mean PMQR

Response: Thanks for the reminder. The text has been corrected.

Line 302 - the small/absent effect of aac gene, at least, is well documented in many other studies - including those testing a wide variety of Salmonella serotypes and using WGS in comparison to the phenotype

Lines 302-303 - also the additive contribution - mainly in addition to chromosomal mutations was well documented

Response: Thank you for your comment. Although the small effect of aac gene is well documented in other studies, our main focus was the synergistic effect between oqxAB and aac.

Figure 1 - which statistical tests and which comparisons should be stated clearly in the methods and not only here

Response: Thank you for your comment. The method of statistical test has been added to the caption for figure 1 and methodology section.

Line 376 - relative - what was the reference? pAAC?

Response: The WT ABRp was used as the reference.

Figure 2 - please clarify the split in TSS - for example - what is the meaning and how it was determined that the base in location 199 is part of the TSS of pHK0653 and not for example a base in location 196?or 180?

Response: Thank you for your question. The transcription start site analysis was done by 5'-race system and the purified gene product was sent to Sanger sequencing for extracting sequence of the transcription start site. The TSS of pHK0653 was the result we got from Sanger sequencing but not a prediction on our own.

March 5, 2022

Dr. Sheng Chen
CityU of Hong Kong
Department of Infectious Diseases and Public Health
Department of PH
Kowloon
Hong Kong

Re: Spectrum02170-21R1 (Transcriptional regulation and functional characterization of the plasmid-borne *oqxAB* genes in *Salmonella Typhimurium*)

Dear Dr. Sheng Chen:

Your manuscript has been accepted, and I am forwarding it to the ASM Journals Department for publication. You will be notified when your proofs are ready to be viewed.

Sincerely,

Sandeep Tamber
Editor, Microbiology Spectrum

Journals Department
Supplemental Dataset: Accept